# Stability Enhancement of Freeze-Dried Gelatin/Alginate Coacervates for bFGF Delivery

**DOI:** 10.3390/pharmaceutics14122548

**Published:** 2022-11-22

**Authors:** JongOk Lee, Eunmi Ban, Heejung Park, Aeri Kim

**Affiliations:** Department of Pharmacy, College of Pharmacy, CHA University, Seongnam-si 463-400, Republic of Korea

**Keywords:** diabetic foot ulcer, coacervates, fibroblast growth factor, freeze-drying, cryoprotectants

## Abstract

Chronic wound sites have elevated levels of proteolytic enzymes that negate the activity of topically applied growth factors. bFGF encapsulated in gelatin/alginate coacervates was protected from protease and showed better activity than bFGF in solution; however, its activity decreased with particle size and PDI increase after freeze-drying and rehydration. In this study, we aim to improve the stability of bFGF coacervates during freeze-drying to enable a topically applied growth factor delivery system for diabetic foot ulcer. Trehalose, mannitol, and Tween 80 at various concentrations were tested as cryoprotectant candidates. Trehalose improved the mechanical property of freeze-dried coacervates and physical properties after rehydration, resulting in stable size and PDI values. It also enhanced the bFGF activity in hyperglycemic human dermal fibroblasts with better cell viability, migration, and procollagen synthesis compared to the coacervates without trehalose. Hydrogen bonding interactions between trehalose and polymers probed by ATR-FTIR contribute to the stability of coacervates during freeze-drying. In conclusion, the freeze-dried gelatin/alginate coacervates encapsulating bFGF was effectively stabilized with trehalose, and the resulting coacervate composition is suggested as a potential therapeutic modality for chronic wounds including diabetic foot ulcer.

## 1. Introduction

Diabetic foot ulcer(DFU) is a type of chronic complication that occurs in about 15–25% of diabetic patients. It is steadily increasing as the number of diabetic patients increases worldwide [1,2]. Basic fibroblast growth factor (bFGF) is known to be effective in treating wounds, including DFU because it helps form granulation tissues by stimulating angiogenesis and proliferation of fibroblast [3,4]. However, the chronic wound sites are known to have elevated levels of proteolytic enzymes and peptidases, thus negating the effect of topically applied growth factors [5]. Various formulations such as nanofibers, gels, microspheres, and nanoparticles have been investigated to protect bFGF from these proteolytic enzymes [6,7]. Bioactive agent-loaded electrospun nanofiber membranes have also gained attention in wound healing applications [8].

In our previous studies, gelatin/alginate(GA/SA) coacervates were used as a topical delivery system of growth factors for the treatment of DFU [9,10]. Gelatin and alginate are well-known biodegradable and biocompatible polymers effective in wound healing [11]. These two types of polymers have different net charges depending on the pH of the reaction media and can encapsulate bFGF through electrostatic interactions. The bFGF encapsulated in GA/SA coacervates was effectively protected from protease and showed a better effect on wound healing [10].

Coacervates in a liquid state can be freeze-dried for enhanced stability and long-term storage or accurate unit dosing [12,13]. Freeze-drying can be an attractive process to preserve the complex structures of biologically active materials [14]. However, some problems were reported during the freezing and drying process. During freezing, water molecules form ice crystal structure, and the polymers can shrink as the ice volume increases [15,16]. During drying steps, deformation of the polymer matrix may occur [17,18]. As a result, freeze-drying can lead to polymer aggregation, leakage of encapsulated growth factors, and damage. Many studies have stabilized the formulation by adding a cryoprotectant before freeze-drying [19,20].

The stability of freeze-dried GA/SA coacervates or changes after rehydration of freeze-dried coacervates were not examined in our previous study [10]. In this study, we aimed to improve the shape and texture of lyophilized GA/SA coacervates and the bFGF stability by adding an appropriate cryoprotectant, thereby reducing the physical stresses to the GA/SA coacervates during the freeze-drying process.

## 2. Materials and Methods

### 2.1. Materials

Gel strength 300 gelatin A (HWGA, average 50–100 kDa) and acetic acid were purchased from Dae Jung (Gyeonggi-do, Republic of Korea). Alginic acid sodium salt from brown algae(SA, low viscosity) and citric acid were purchased from Sigma–Aldrich (St. Louis, MO, USA). FGF (bFGF) was provided by CHA meditech (Gyeonggi-do, Republic of Korea). D-(+)- trehalose dihydrate, D-mannitol and TWEEN^®^ 80 were purchased from Sigma–Aldrich (St. Louis, MO, USA). Fluorescein-5-Isothiocyanate (FITC) purchased from TCI (Tokyo, Japan). bFGF ELISA kit was purchased from R&D Systems (Minneapolis, MN, USA).

Basic Dulbecco’s phosphate buffered saline (DPBS) was purchased from Welgene (Gyeongsan-si, Republic of Korea), Dulbecco’s Modified Eagle Medium (DMEM) was obtained from Hyclone (Logan, UT, USA), 0.25% Trypsin-EDTA and fetal bovine serum (FBS) were purchased from Gibco (Paisley, UK). Neonate HDF cells(Cat. no. CC-2509, NHDF-Neo) were from Lonza (Walkersville, MD, USA). The cell counting kit-8 (CCK-8) was obtained from Dojindo (Kumamoto, Japan). Mitomycin C from Streptomyces caespitosus was purchased from Sigma–Aldrich (St Louis, MO, USA). Procollagen I C-Terminal propeptide (PICP) ELISA kit was purchased from Aviva systems biology (San Diego, CA, USA).

### 2.2. Fabrication of Gelatin/Alginate Coacervates with or without Cryoprotectant

1% HWGA (*w/w*) and 1% SA (*w/w*) solutions in distilled water were mixed at 1:1 ratio. 0.5 M acetic acid or 0.1 M citric acid were added to the mixed solutions of GA/SA to adjust pH 4.34 to prepare acetic acid coacervates or citric acid coacervates, respectively. For encapsulation of bFGF, a bFGF solution was added into acidified GA solutions before adding the SA solution. The concentration of bFGF or FITC-labeled bFGF (FITC-bFGF) varied depending on the experimental purpose, as described in each method. Trehalose, mannitol, and Tween 80 were tested as cryoprotectants. Each of these cryoprotectant stock solutions in distilled water (100 mg/mL) was added to the coacervates at the final concentration of 1, 2, or 3%. 

### 2.3. Freeze-Drying Process of Complex Coacervates

Depending on the purpose of the experiments, 100 µL coacervates in 96 well-plates or 1 mL coacervates in Eppendorf tube were freeze-dried in a freeze-dryer (alpha 1-2 LD plus, Martin Christ, Germany) after freezing overnight in a −80 °C deep freezer. Primary drying was performed at −30 °C and 0.37 mbar for 15 h, and secondary drying was performed at −80 °C and 0.001 mbar for 2 h. The freeze-dried samples were kept at −20 °C in heat-sealed aluminum pouches until use. 

### 2.4. Characterization of Coacervates

#### 2.4.1. Coacervates before Freeze-Drying and after Rehydration of Freeze-Dried Coacervates

Coacervates were visually examined in cuvettes, and their pictures were taken. Encapsulation efficiency of bFGF in coacervates was determined as previously described [10]. Freeze-dried coacervates were rehydrated with distilled water to make the same final volume as the coacervate before freeze-drying. Coacervates without cryoprotectants and those with trehalose, mannitol, and Tween 80 at distinct concentrations were compared. The sample’s particle size and PDI values were measured by dynamic light scattering (Zetasizer Nano ZS, Malvern Instruments Ltd., Almelo, UK) with back-scatter detection at a scattering angle of 173°. The samples in polystyrene disposable cuvettes were allowed to equilibrate at 25 °C before the measurement. 

For fluorescence microscopic examination, 10 µg of FITC-labeled bFGF was encapsulated in the coacervates. The FITC conjugation method was followed by Sigma-Aldrich protocol. Briefly, 50 µL of FITC solution (1 mg/mL in dimethylsulfoxide) was added slowly into 1 mL bFGF solution (2 mg/mL in bicarbonate buffer, pH = 9.2, 0.1 M). The mixture was incubated for 8 h at 4 °C in darkness gently shaking. FITC-labeled bFGF (FITC-bFGF) was separated from the FITC solution by a Sephadex G-50 (Cytiva) column (Φ24 × h400 mm). Sephadex G-50 beads 1.5 g were equilibrated with Phosphate buffer saline 15 mL for 1 h. The fluorescence of coacervates encapsulating FITC-bFGF was monitored using Axio 5 fluorescence microscope (Carl Zeiss, Germany) at 495 nm excitation, 525 nm emission.

#### 2.4.2. Characteristics of Freeze-Dried Coacervates 

##### Scanning Electron Microscope (SEM) Analysis

Images of freeze-dried coacervates with or without trehalose 2% (*w/v*) were taken by a scanning electron microscope (SNE-4500 M, SEC Co., Ltd., Suwon, Republic of Korea). Adhesive carbon tape was attached to stainless steel stubs. The samples were placed on the stub and coated with platinum using Magnetron coating M/C (MCM-100). The pieces were then placed in the chamber of the SEM and monitored at 100×, 400×, and 1000× magnifications.

##### ATR-FTIR

FT-IR spectra of freeze-dried samples were obtained with an FT-IR (Shimadzu, IRSpirit, Japan) with ATR accessory (PIKE technologies, WI, USA). Freeze-dried coacervates were loaded onto the accessory plate for analysis. A total 100 scans were collected in the range of wavelength 4000–650 cm^−1^.

##### Thermal Analysis

Thermogravimetric analysis (TGA) of samples was performed with the TA-Q50 instrument (TA Instruments, USA) to determine the water content of the freeze-dried coacervates. Samples were loaded into a platinum pan and heated at a rate of 10 °C/min under nitrogen purging gas at the rate of 50 mL/min. Differential scanning calorimetry (DSC) measurements of the samples were performed using the TA-Q20 instrument (TA Instruments, USA). Samples were placed in Tzero Pan and Lid under nitrogen purging gas at the rate of 50 mL/min. Heating-cooling-heating cycles or heating-10 min stay at 100 °C-cooling-heating cycles were performed.

##### bFGF Stability Measurement

The bFGF ELISA was conducted with rehydrated coacervates, freeze-dried with or without trehalose (2%, *w/v*), to assess the bFGF stability in GA/SA coacervates. Coacervates were fabricated as described above in the Section 2.2 and Section 2.3. After freeze drying, coacervates were rehydrated in PBS to release the bFGF encapsulated in coacervates. An ELISA assay was carried out according to the bFGF ELISA Kit protocol.

##### Release Test

In vitro release tests of freeze-dried coacervates with or without trehalose (2% *w/v*) were conducted using the Transwell method described in previous studies [10]. Freeze-dried coacervates with a total polymer weight of 4 mg and a GA to SA ratio of 1:1, encapsulating 10 µg of FITC-bFGF, were put into the Transwell inserts, and 100 µL DW was added for rehydration of the samples. A release medium (900 µL DPBS) was placed in the wells of the 24-well plate as the receiver phase. As a control sample, a 100 µL of FITC-bFGF solution (100 µg/mL) was placed in the Transwell insert. The samples were incubated in a shaking incubator at 32 °C and 50 rpm. Transwell inserts containing samples were moved to the next well containing fresh DPBS at 1, 2, 4, and 8 h to collect the released FITC-bFGF at each time point. The fluorescent intensity in each well was measured at 495 nm for excitation and 525 nm for emission in the SynergyMx microplate reader (Biotek Instruments, Santa Clara, CA, USA).

#### 2.4.3. In Vitro Cell Activity Study

##### HDFs Viability Test

Freeze-dried bFGF coacervates were prepared with a 1:1 ratio of HWGA/SA (4 mg) and 10 ng of bFGF with 0.00025% drug loading (10 ng/4 mg). Human dermal fibroblasts (HDFs) were cultured in DMEM/low glucose supplemented with 10% FBS, penicillin (100 U/mL), and streptomycin (100 µg/mL) in a 5% CO_2_ incubator at 37 °C. For hyperglycemic conditions, high glucose DMEM (25 mM) was used (Hyclone, Logan, UT, USA). Cells were cultured at passage 3∼6 in a 100 mm dish until 80% confluence, then 1 × 10^4^ cells per well were seeded in a 24-well plate with low glucose media (5 mM glucose) conditioning media and were incubated at 37 °C for 24 h. After that, each well was washed once with DPBS and replenished with a media containing 0.5% FBS in each medium except for the Positive group (10% FBS) and the bFGF group (10 ng/mL) bFGF in 1 mL of 0.5% FBS High glucose DMEM. Freeze-dried samples were put in the Transwell inserts, and 100 µL of the high glucose DMEM containing 0.5% FBS was added to rehydrate the samples. After these inserts were assembled into each well, the cells were incubated at 37 °C and 5% CO_2_ for 5 days. The HDF viability was determined by the CCK-8 assay. All results were obtained from measurements at 450 nm and normalized to the results of the negative group (0.5% FBS, without treated sample). A separate set of experiments was done without FBS in the culture media. The supernatant of those samples was used for the PICP assay described below.

##### HDFs Procollagen Synthesis

For the PICP experiment, cells were cultured in the same method as the viability test except that the culture media did not contain FBS in order to minimize the ELISA interference by FBS (Xiao and Isaacs 2012), and the supernatant of the cultured media was collected after 5 days incubation. An ELISA assay was carried out following the PICP ELISA Kit protocol.

##### HDFs Scratch Wound Assay

After marking a mid-line at a 24-well bottom with a marker pen, each well was coated with 0.1% gelatin. The plates were incubated in a 37 °C incubator for 2 h and washed twice with DPBS. HDFs were cultured under normal conditions at Passage 3~6 in a 100 mm dish until 80% confluence. After cells were detached using 0.25% Trypsin-EDTA, 1 × 10^5^ cells per well were incubated at 37 °C and 5% CO_2_ overnight. After that, autoclaved 1 mL pipette tips were used to scratch the cells along the mid-line of the well, and the wells were washed twice with DPBS to remove cell debris. Cells were treated with 1 mL of media containing 0.5% FBS and 10 µg/mL of mitomycin C and incubated at 37 °C for 2 h. After that, images were taken with a light microscope (Leica, Wetzlar, Germany) at each time point after scratching. Freeze-dried samples were put in the Transwell inserts, and 100 µL of media containing 0.5% FBS in DMEM/high glucose was added to the inserts. The incubation media (900 µL) contained 0.5% FBS, except for the Positive (10% FBS, 1 mL). After the cells were incubated at 37 °C for 24 h, the wound closure width was evaluated using Image J (National Institutes of Health, Bethesda, MD, USA). The wound-healing effects of various samples were normalized to that of the Negative group as shown in the following equation.
Fold of wound area = {(A0 − A24)/A0} sample/{(A0 − A24)/A0} Negative
where A0 is the wound area at 0 h and A24 is area of the wound after 24 h.

### 2.5. Statistical Analysis

Data are expressed as the mean ± standard deviation (SD). Statistical analysis was performed using one-way analysis of variance (one-way ANOVA) followed by Dunnett’s test with GraphPad Prism 5 software (version 5.01, GraphPad Software Inc., La Jolla, CA, USA). For all tests, *** *p* < 0.001, ** *p* < 0.01, * *p* < 0.05.

## 3. Results

### 3.1. Characterization of Gelatin/Alginate Coacervates

Table 1 shows the characteristics of a representative liquid state coacervate.

### 3.2. Effects of Cryoprotectants on Characteristics of Rehydrated Coacervates

#### 3.2.1. Effect on Aggregation and Turbidity Changes upon Rehydraion after Freeze-Drying

In the case of freeze-dried coacervates without cryoprotectant, rehydration with an equal volume of D.W showed aggregation (Figure 1a). Addition of trehalose or mannitol resulted in homogeneous phase without aggregation and turbidity decrease. However, the addition of Tween 80 caused a significant decrease in turbidity at 1% and 2% (*w/v*) (Figure 1b,c).

#### 3.2.2. Effects on Particle Size and PDI Value

After rehydration of freeze-dried coacervates, the changes in size and PDI values differed depending on the acidifying agent and addition of cryoprotectant. The size and PDI values of AA-F/D and CA-F/D coacervates without cryoprotectants increased due to aggregation after rehydration (Figure 2). CA-F/D showed larger increases in both size and PDI than AA-F/D. In contrast, the size and PDI values returned to the original values after rehydration of CAT-F/D (2% and 3% trehalose, *w/v*) and CAM-F/D (1, 2, and 3% mannitol, *w/v*) (Figure 3). However, in the case of Tween 80, the size and PDI values increased at all concentrations.

#### 3.2.3. Fluorescence Microscope Images

Fluorescence microscope images of coacervate encapsulating FITC labeled bFGF are shown in Figure 4 and Figure 5. The liquid state FITC-bFGF coacervates before freeze-drying show evenly distributed fluorescence of coacervates without aggregation (Figure 4a and Figure 5a). Aggregation in rehydrated samples of freeze-dried coacervates was more extensive in citric acid coacervates than acetic acid coacervates (Figure 4b and Figure 5b). When trehalose or mannitol was added, rehydrated samples returned to similar to the liquid state coacervate before freeze-drying (Figure 4c,d and Figure 5c,d). In the case of Tween 80, extensive aggregation appeared (Figure 4e and Figure 5e).

### 3.3. Effects of Cryoprotectants on Characteristics of Freeze-Dried Coacervates

#### 3.3.1. Morphology and Texture of Freeze-Dried Coacervates

Coacervates for DFU will be administered topically by placing freeze-dried samples on wound sites. Therefore, physical integrity is important for proper application. However, the freeze-dried sample of CAM-F/D was quickly broken and could not be held with a tweezer. Samples containing Tween80 appeared contracted. The CAT-F/D was easier to handle than the other groups (Figure 6a). Therefore, 2% (*w/v*) trehalose was finally selected for further evaluation. Subsequent experiments were performed with coacervates containing 2% (*w/v*) trehalose.

#### 3.3.2. SEM Images of Freeze-Dried Coacervates

Interconnected sheets were observed in SEM images of freeze-dried coacervates (Figure 7). At the same microscopic scale, samples without trehalose showed much larger pores than AAT-F/D and CAT-F/D groups.

#### 3.3.3. ATR-FTIR Spectra of GA/SA Coacervate with Trehalose

FTIR was used to examine the intermolecular interactions between trehalose and components of coacervates (Figure 8). The spectra of the powder mixture overlap with those of trehalose because of the relatively high ratio of trehalose in the powder mixture. Therefore, the interaction between trehalose and coacervate was monitored by the changes in the trehalose peaks. FTIR spectra of trehalose previously reported have been used for peak identification [21,22]. Table 2 shows the main peaks of trehalose, two vibrational modes (asymmetrical and symmetrical) of the α,α-1↔1-glycosidic bond at 994, 956 cm^−1^, and C-H bonds in alpha anomers at 841, 851 cm^−1^. While there was no change of these peaks in the physical mixtures, these bands shifted upon freeze-drying due to the hydrogen bonding interactions between polymers and trehalose. In addition, C1-H, CH_2_, and C-O-H band of trehalose at 910 cm^−1^ disappeared.

#### 3.3.4. bFGF Stability after Freeze-Drying Process

bFGF ELISA was performed to examine the effect of cryoprotectant on the stability of bFGF encapsulated in the coacervates (Figure 9). When trehalose was added to acetic acid and citric acid coacervates, the stability of bFGF was increased compared to those without trehalose.

#### 3.3.5. Controlled Release Rate of Freeze-Dried Coacervates with Trehalose

Figure 10 compares the release rate of bFGF solution, a physical mixture, and various coacervates. The statistical differences in each time point is provided in Table 3. In the case of AA-F/D, AAT-F/D, and Physical mixture-F/D, 63.3 ± 1.8%, 71.2 ± 1.9%, and 65.7 ± 3.4% were released in 4 h, respectively. These are not significantly different from 77.4 ± 1.6% of the bFGF solution group. CA-F/D showed the lowest release rate due to aggregation at 4 h (19.7 ± 1.2%), whereas CAT-F/D showed the release rate between CA-F/D and other groups at 4 h (38.6 ± 1.3%). At 8 h, 100% was released in all groups except CA-F/D and CAT-F/D.

### 3.4. In Vitro Cell Activity

#### 3.4.1. In Vitro HDFs Viability Assay

The effects on HDF viability were tested by CCK-8 assay under hyperglycemic condition (Figure 11). The cell culture conditions are provided in Table 4. All the coacervate groups showed improved activity compared to the negative control and bFGF solution groups. However, compared to AA and CA groups which were liquid state before freeze-drying, rehydrated AA-F/D and CA-F/D showed decreased activity (Figure 11a). AAT-F/D and CAT-F/D groups showed enhanced activity compared to those without trehalose group (Figure 11b).

#### 3.4.2. In Vitro HDF Procollagen Synthesis

The secretion of procollagen Type I C-Peptide (PICP), an indicator of collagen production, plays an important role in skin regeneration at the wound site (Nickel, Wensorra et al., 2021). All the sample groups showed improved activity compared to the negative group, and especially the CAT-F/D group to which trehalose was added showed the highest PICP synthesis at day 5 incubation (Figure 12).

#### 3.4.3. In Vitro HDF Scratch Wound Assay

Figure 13a shows representative images of HDF under the hyperglycemic condition at 0 and 24 h treatment after scratching. Figure 13b shows wound healing rates normalized to the Negative group. The Positive group with 10% FBS showed higher activity than the Negative group with 0.5% FBS. The AA-F/D, CA-F/D and AAT-F/D groups showed improved activity compared to the negative group, but not as much as the Positive group. However, CAT-F/D group showed a similar wound healing effect compared the Positive group (Positive group: 1.78 ± 0.04-fold, CAT-F/D group: 1.92 ± 0.05-fold, normalized to the negative group).

## 4. Discussion

In our previous study, the effects of bFGF encapsulated in GA/SA coacervates were evaluated by in vitro activity studies, including HDF proliferation, migration and procollagen synthesis under hyperglycemic condition. GA/SA coacervate system was fabricated with an optimized composition using DOE, to achieve stable size, PDI values without precipitation [10]. However, aggregation appeared when the freeze-dried coacervates were rehydrated (Figure 1). Such changes can be attributed to the stress that occurs during freeze-drying process [14]. Our hypothesis was that an appropriate cryoprotectant would prevent polymer aggregation and drug leakage thereby maintaining the original properties of the coacervates.

The changes in size and PDI values of rehydrated samples after freeze-drying differed depending on the acidifying agent: rehydrated CA-F/D showed larger increases in the size and PDI than rehydrated AA-F/D (Figure 2). As shown in our previous study, GA/SA coacervation occurs in a narrow pH range [10]. The pH of rehydrated AA-F/D and CA-F/D were 5.1 and 4.43, respectively. The pH of AA-F/D after rehydration increased compared to that measured at the time of coacervate preparation. One may speculate that some acetic acid volatilizes during the freeze-drying phase, and the pH of rehydrated samples rises, thus reducing the electrostatic attraction between gelatin and alginate. This will in turn result in less coacervation and aggregation. On the other hand, there was no change in the pH of CA-F/D, leading to aggregation from freeze-drying stress. Fluorescence microscopy of coacervates encapsulating FITC-labeled bFGF demonstrated aggregation after rehydration of freeze-dried coacervates (Figure 5).

Since GA/SA coacervates have pH-dependent characteristics, cryoprotectants that affect the pH are inappropriate. Therefore, we have tested trehalose, mannitol, and Tween 80 as candidate cryoprotectants. These are widely used cryoprotectants in many studies and currently marketed products [23]. Trehalose and mannitol prevented aggregation during hydration of freeze-dried coacervates (Figure 3). CAT-F/D returned to the original size and PDI value upon rehydration after freeze-drying. Mechanical properties of freeze-dried coacervates also differed depending on the cryoprotectant. Trehalose gave better mechanical properties than mannitol or Tween 80 in freeze-dried GA/SA coacervates (Figure 6). SEM images of the freeze-dried GA/SA coacervates without trehalose exhibited large pores. In contrast, those of samples containing 2% (*w/v*) trehalose showed much smaller size pores in the interconnected sheets of polymers (Figure 7). This finding demonstrates that trehalose prevents large ice crystal growth during freezing process which would help to maintain the polymer networks in the coacervates. The changes in the trehalose peaks in FT-IR indicates that trehalose bound to the components of coacervates replace the sublimated water molecules and help to maintain the polymer matrix (Figure 8). The water content of freeze-dried coacervates were determined using TGA. Water content of samples with trehalose was less than that of samples without trehalose (1.77 ± 1.42 vs. 5.4 ± 2.48, *n* = 2) (Appendix A). The result is consistent with the findings by Neupane et al.^14^. We speculate that the intermolecular interactions between trehalose and polymers displace water molecules held by polymers, facilitating sublimation of water vapor during drying cycle. DSC profiles of HWGA, SA, coacervates without and with trehaloose are shown in Appendix A. The first heating runs of all samples showed large endortherms which were not observed during second heating after cooling. These endotherms could be due to dehydration or enthalpy relaxation. DSC profiles of heating-cooling-heating cycles were similar (Appendix A). However, the second heating runs after heating-10 min stay at 100 °C-cooling-heating cycles differed in their slope (Appendix A): the sample with trehalose showed descending slope around Tg, whereas that without trehalose showed ascending slope around Tg. The glass transition temperature of these samples were not significantly different. Further study is necessary to interpret such differences in the thermal behavior around Tg. The addition of trehalose also enhanced the stability of bFGF by reducing freeze-drying stresses (Figure 9).

Controlling the release rate of growth factors is an important factor for successful wound healing. bFGF has a short half-life and is readily degraded by proteases. Also, burst release patterns can cause undesirable systemic exposure, which might lead to adverse events. Therefore, it is crucial to have an appropriate release rate [24]. AA-F/D and AAT-F/D showed similar release rates as bFGF Solution. As discussed above, it can be attributed to the loss of some acetic acid during the freeze-drying phase and the subsequent pH increase. Their release rate is similar to the bFGF solution because the pH is not within the coacervation range any more. CA-F/D showed the lowest release rate due to aggregation. CAT-F/D showed the release rate between those of AA-F/D and CA-F/D. CAT-F/D showed no aggregation and maintained the coacervation pH (pH4.43) unlike acetic acid coacervate (Figure 10). These results highlight the importance of an appropriate acidifying agent and cryoprotectant in maintaining the characteristics of coacervates during freeze-drying.

The bFGF solution did not revitalize the HDF in 0.5% FBS under hyperglycemic condition (Figure 11a). Coacervate groups showed improved HDF viability compared to the Negative control and bFGF solution groups. However, activities of AA-F/D and CA-F/D were significantly less than those of AA and CA. These differences are consistent with those observed in ELISA (Figure 9). These results indicate and the bFGF solution is not an effective way to delivery bFGF for hyperglycemic wound healing and the freeze-drying process of coacervates negatively affect the bFGF activity. In particular, HDF migration in the scratch wound assay is an important factor in predicting successful skin regeneration during the wound healing phase [25]. In HDF cell activity tests including viability, migration and procollagen synthesis, AAT-F/D and CAT-F/D showed the higher activity than samples without trehalose (Figure 11, Figure 12 and Figure 13). The best activity in all three cellular activity tests was observed in the CAT-F/D group, which showed a controlled release pattern between the CA-F/D group and those of the AA-F/D, AAT-F/D, and PM groups.

Further study is necessary to determine whether such slower release rate and better in vitro activity of CAT-F/D will be translated to better in vivo efficacy. In addition, stability studies on long-term storage of CAT-F/D are needed.

## 5. Conclusions

Coacervation of GA/SA depends on the pH of the reaction mixture, and therefore an acidifying agent is required. Citric acid was a more appropriate acidifying agent than acetic acid because of its non-volatility during the freeze-drying process. With citric acid, the pH of rehydrated GA/SA samples and their characteristics after freeze-drying remained the same as before the freeze-drying. We selected trehalose as a cryoprotectant suitable for GA/SA coacervates to mitigate stress during freeze-drying. Intermolecular interactions between trehalose and GA/SA prevented the formation of large ice crystals during freezing, which led to the stabilization of freeze-dried coacervates. Adding trehalose in GA/SA coacervates also improved the stability of encapsulated bFGF, and thereby enhanced its activity on the viability, migration, and procollagen synthesis of HDF. In conclusion, we propose CAT-F/D as a potential therapeutic modality for DFU. Further studies on in vivo efficacy and product stability are warranted to translate the current findings to clinics.

## Figures and Tables

**Figure 1 pharmaceutics-14-02548-f001:**
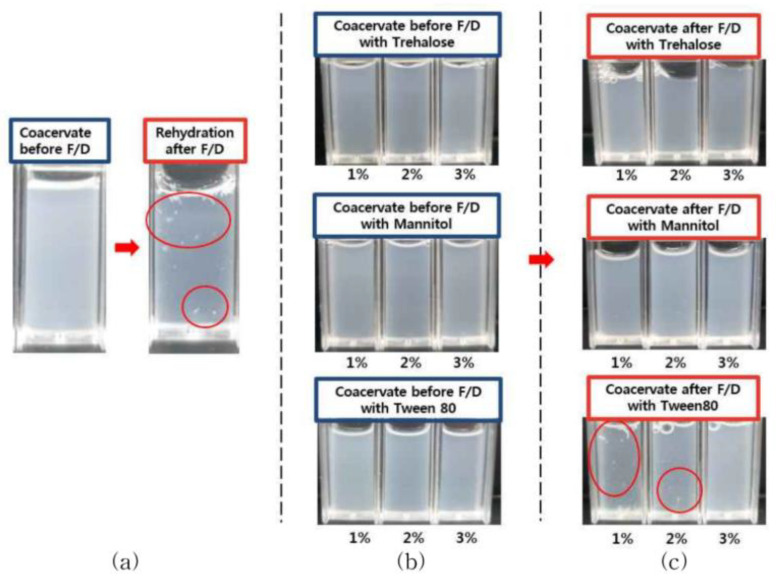
Appearance of GA/SA coacervates acidified with citric acid before freeze-drying (F/D) and after rehydration of freeze-dried samples; (**a**) without cryoprotectant, (**b**) coacervates containing trehalose, mannitol, or Tween80 before freeze-drying, (**c**) after rehydration of freeze-dried coacervates containing trehalose, mannitol, or Tween80. Red circles show aggregated particles.

**Figure 2 pharmaceutics-14-02548-f002:**
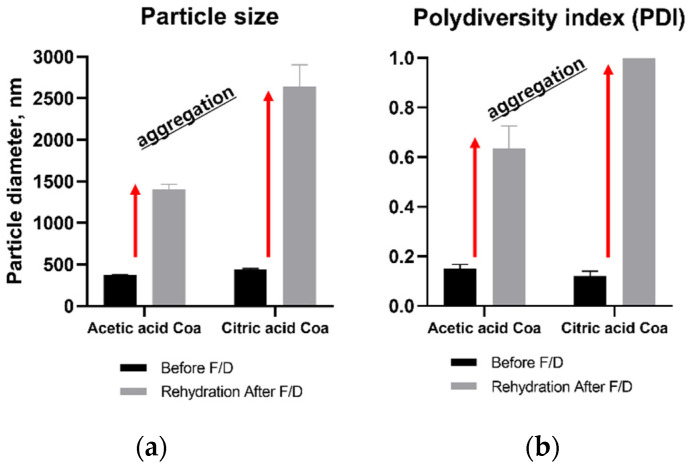
Effects of freeze-drying (F/D) on GA/SA coacervates acidified with acetic acid or citric acid; (**a**) particle size, (**b**) PDI before freeze drying and after rehydration of freeze-dried coacervates.

**Figure 3 pharmaceutics-14-02548-f003:**
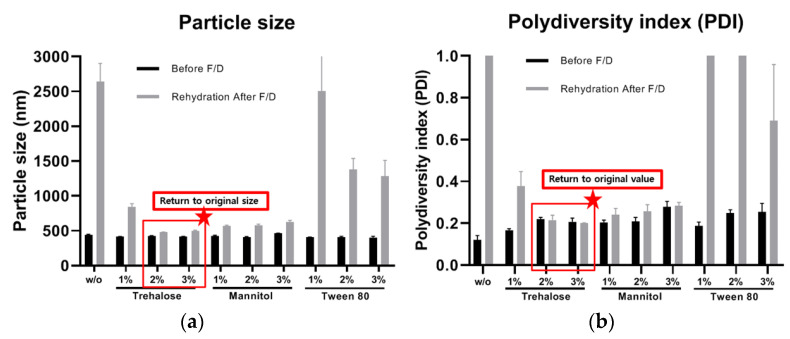
Effects of freeze-drying (F/D) on GA/SA coacervates acidified with citric acid without (*w/o*) or with trehalose, mannitol, and Tween 80; (**a**) particle size and (**b**) PDI value before freeze drying (F/D) and after rehydration of freeze-dried coacervates.

**Figure 4 pharmaceutics-14-02548-f004:**
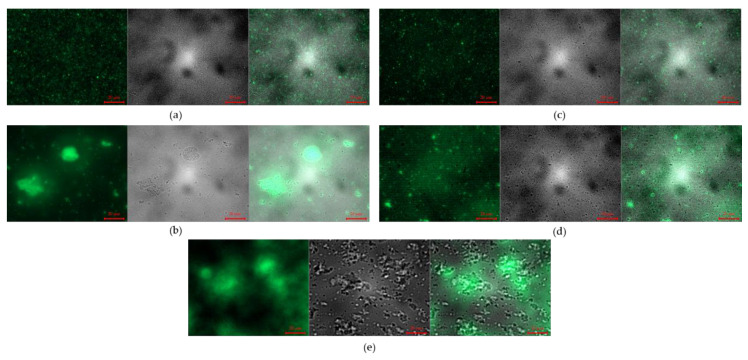
Microscope images of GA/SA coacervates acidified with citric acid without or with cryoprotectants (Left: Fluorescence images, Middle: Optical images, Right: Overlapped Fluorescence and Optical images): (**a**) before freeze-drying, (**b**) after rehydration of freeze-dried coacervates without cryoprotectant, (**c**–**e**) after rehydration of freeze-dried coacervates with trehalose (2%, *w/v*), with mannitol (2%, *w/v*), and with Tween 80 (2%, *w/v*). The scale bars represent 20 μm.

**Figure 5 pharmaceutics-14-02548-f005:**
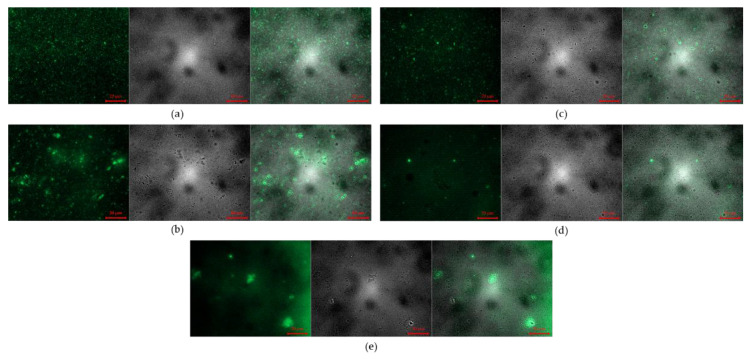
Microscope images of GA/SA coacervates acidified with acetic acid without or with cryoprotectants (Left: Fluorescence images, Middle: Optical images, Right: Overlapped Fluorescence and Optical images): (**a**) before freeze-drying, (**b**) after rehydration of freeze-dried coacervates without cryoprotectant, (**c**–**e**) after rehydration of freeze-dried coacervates with trehalose (2%, *w/v*), with mannitol (2%, *w/v*), and with Tween 80 (2%, *w/v*). The scale bars represent 20 μm.

**Figure 6 pharmaceutics-14-02548-f006:**
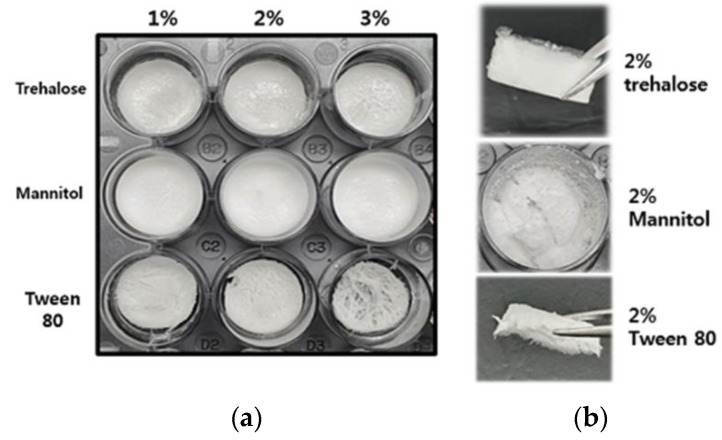
Morphology and texture of freeze-dried GA/SA coacervates acidified with citric acid; (**a**) with various concentrations of cryoprotectants, (**b**) texture of freeze-dried coacervates with 2% (*w/v*) cryoprotectants.

**Figure 7 pharmaceutics-14-02548-f007:**
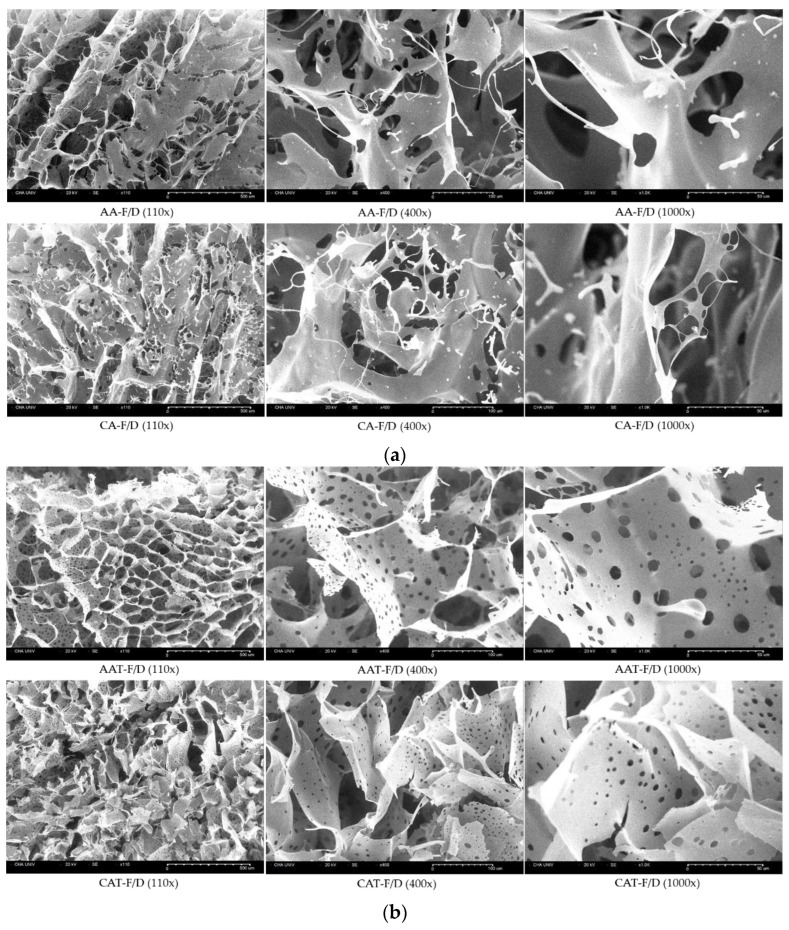
SEM images of freeze-dried GA/SA coacervates acidified with acetic acid or citric acid, (**a**) without trehalose (AA-F/D, CA-F/D) and (**b**) with trehalose (2%, *w/v*) (AAT-F/D, CAT-F/D). Scale bars represent 100 μm, 10 μm, and 10 μm, for 110×, 400×, and 1000×, respectively.

**Figure 8 pharmaceutics-14-02548-f008:**
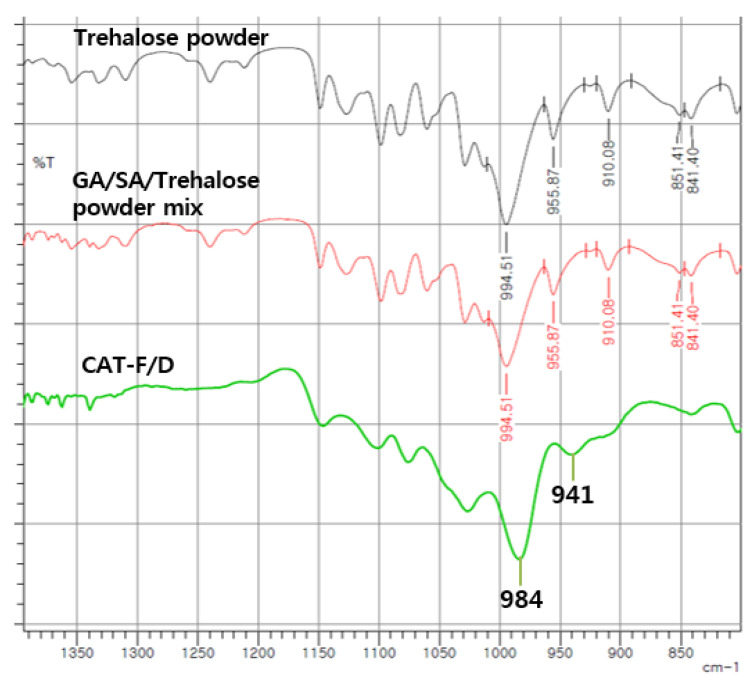
ATR-FTIR spectra of trehalose, GA/SA/trehalose powder mixture and CAT-F/D.

**Figure 9 pharmaceutics-14-02548-f009:**
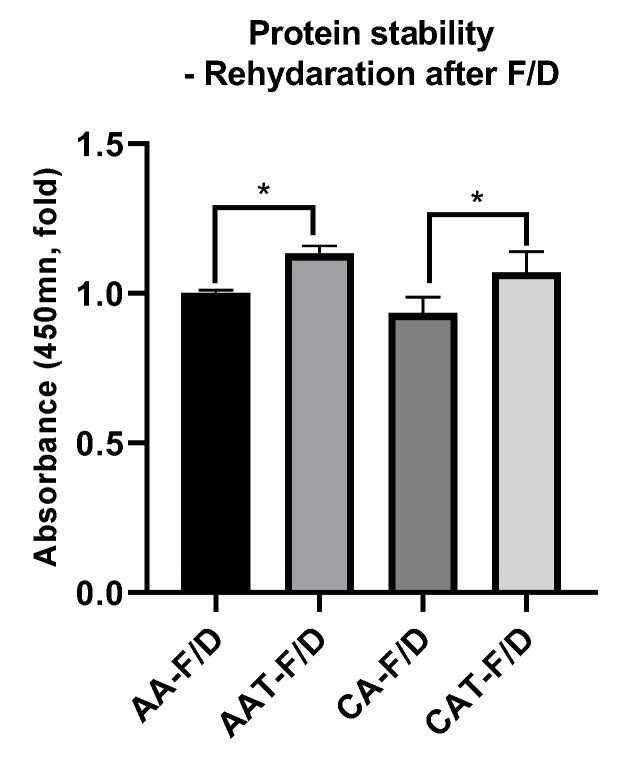
bFGF ELISA results after rehydration of freeze-dried GA/SA coacervates acidified with acetic acid or citric acid, without trehalose (AA-F/D, CA-F/D) or with trehalose (2%, *w/v*) (AAT-F/D, CAT-F/D). * *p* < 0.05.

**Figure 10 pharmaceutics-14-02548-f010:**
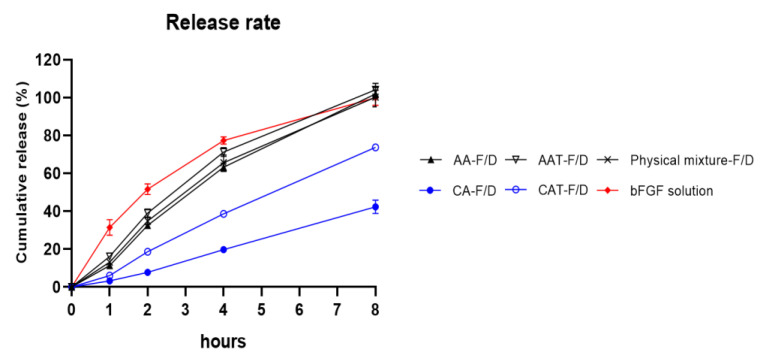
In vitro bFGF release profiles of freeze-dried GA/SA coacervates acidified with acetic acid or citric acid, without trehalose (AA-F/D, CA-F/D) or with trehalose (2%, *w/v*) (AAT-F/D, CAT-F/D), compared to the bFGF solution and freeze-dried physical mixture.

**Figure 11 pharmaceutics-14-02548-f011:**
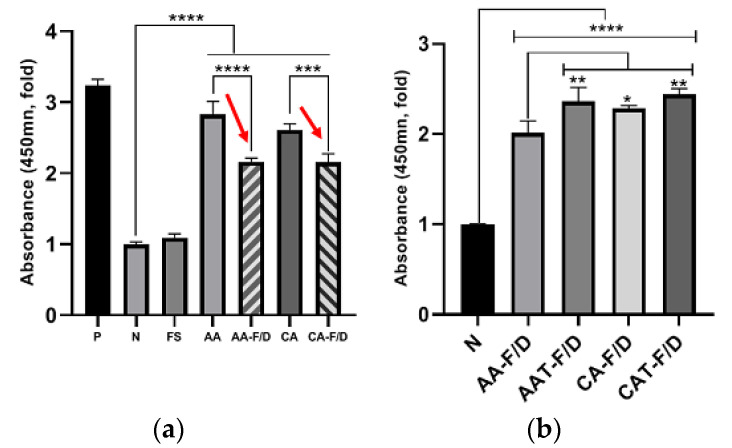
In vitro activity assay of various samples at 5 days (*n* = 3); (**a**) HDF viability after treatment with coacervates before (AA, CA) and after freeze-drying without trehalose (AA-F/D, CA-F/D), (**b**) freeze-dried samples with or without trehalose 2% (*w/v*). P represents the positive control HDF cultured in 5 mM glucose and 10% FBS. Other samples were all cultured in 25 mM glucose and 0.5% FBS. N represents the negative control HDF cultured in 25 mM glucose and 0.5% FBS without any treatment. **** *p* < 0.0001, *** *p* < 0.001, ** *p* < 0.01 * *p* < 0.05. Red arrows emphasize the changes after freeze-drying.

**Figure 12 pharmaceutics-14-02548-f012:**
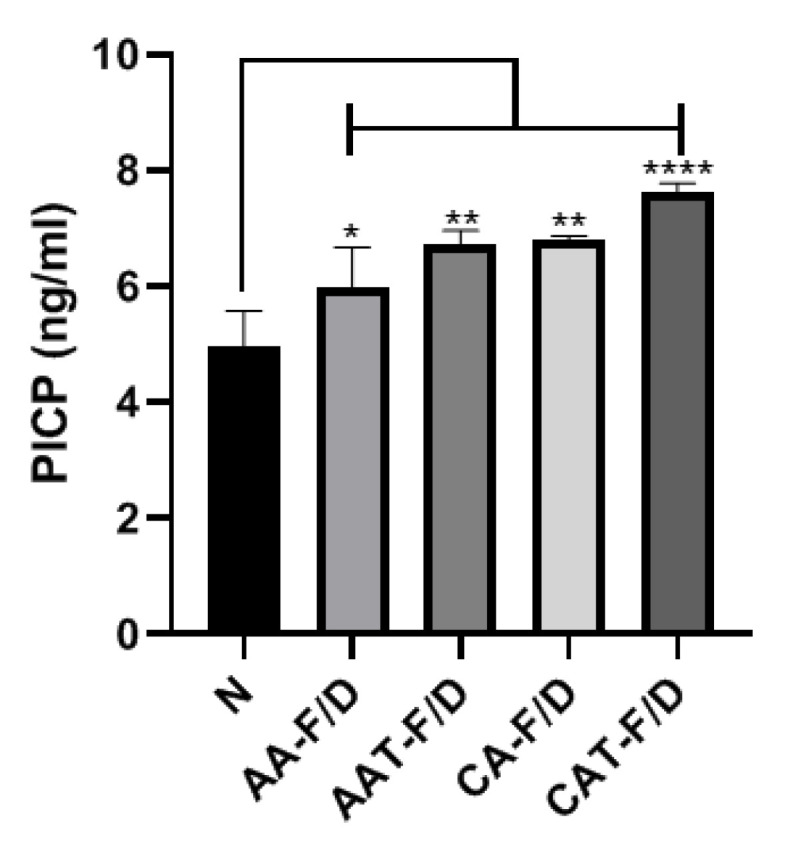
PICP ELISA assay of supernatants various samples collected at day 5. Treatment with freeze-dried samples without (AA-F/D, CA-F/D) or with trehalose 2% (*w/v*) (AAT-F/D, CAT-F/D). Samples were all cultured in 25 mM glucose and 0% FBS, except N, the negative control cultured in 25 mM glucose and 0% FBS without any treatment. **** *p* < 0.0001, ** *p* < 0.01 * *p* < 0.05.

**Figure 13 pharmaceutics-14-02548-f013:**
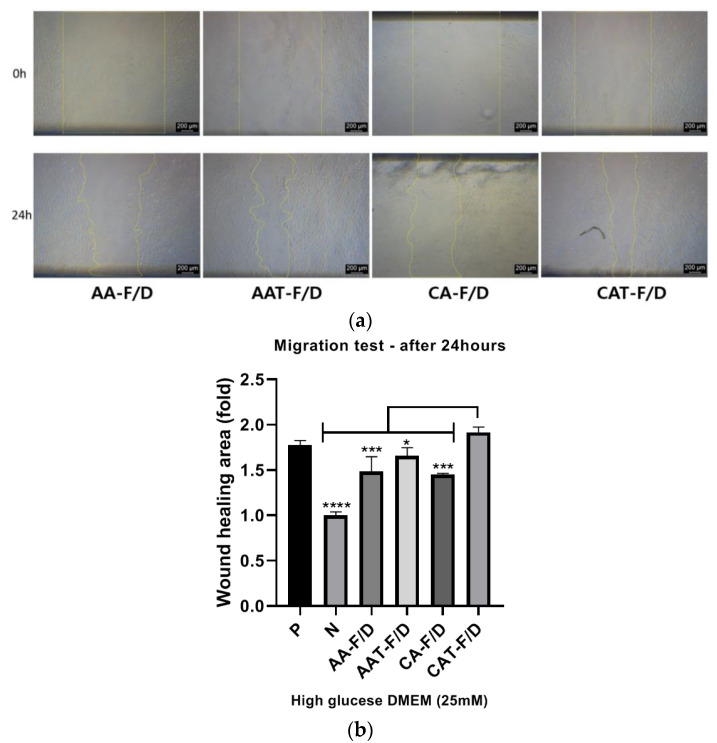
In vitro HDF scratch wound assay results; (**a**) Representative bright field images at right after scratching (0 h) and 24 h after treatment with various samples (24 h), (**b**) Wound healing rates of various conditions relative to the Negative group (*n* = 3). **** *p* < 0.0001, *** *p* < 0.001, * *p* < 0.05.

**Table 1 pharmaceutics-14-02548-t001:** Characteristics of liquid state coacervates encapsulating bFGF (*n* = 3).

HWGA/SA Ratio	Total Polymer Weight	pH	Turbidity	Encapsulation Efficacy%	Z-Average (nm)	PDI
1:1	4 mg	4.34 ± 0.02	1.810 ± 0.01	89.0 ± 0.8	441 ± 8	0.121 ± 0.02

**Table 2 pharmaceutics-14-02548-t002:** ATR-FTIR spectra of trehalose powder, GA/SA/trehalose mixture and CAT-F/D.

Peak Assignment	Trehalose	GA/SA/Trehalose Powder Mixture	CAT-F/D
O-H stretching of trehalose	3274	No change	*
Asymmetrical and symmetrical stretching of the C-H ring	2993, 2973, 2950, 2934, 2907, 2881	No change	*
Two vibrational modes (asymmetrical and symmetrical) of the α,α-1↔1-glycosidic bond	994, 956	No change	994→984, 956→941
Coupled bending vibrations of C1−H, CH_2_ and C−O−H	910	No change	disappeared
Bending vibration of equatorial C−H bonds in α anomers	851, 841	No change	851→disappeared 841→no change

* Peak not detectable due to overlapping peaks in this region originating from trehalose and GA/SA.

**Table 3 pharmaceutics-14-02548-t003:** Statistical difference in the release rate of various samples relative to bFGF solution.

Statistical Analysis					
	AA-F/D	AAT-F/D	CA-F/D	CAT-F/D	Physical
					Mixture
1 h	****	****	****	****	****
2 h	****	****	****	****	****
4 h	****	*	****	****	***
8 h	ns	ns	****	****	ns

**** *p* < 0.0001, *** *p* < 0.001, * *p* < 0.05., ns: not significant

**Table 4 pharmaceutics-14-02548-t004:** Cell culture conditions for each test.

Figure 12	FBS	Media
Negative group	0%	High glucose DMEM
Coacervate group	0%	High glucose DMEM
**Figure 13a,b**	**FBS**	**Media**
Positive group	10% (*w/v*)	High glucose DMEM
Negative group	0.5% (*w/v*)	High glucose DMEM
Coacervate group	0.5% (*w/v*)	High glucose DMEM

## Data Availability

Data is contained within the article.

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
