# Peer review of "Stability Enhancement of Freeze-Dried Gelatin/Alginate Coacervates for bFGF Delivery"

_pharmaceutics, 2022, doi:10.3390/pharmaceutics14122548_

Round 1

Reviewer 1 Report

1. The study presents the results of original research.

2. Results reported have not been published elsewhere.

3. Experiments, statistics, and other analyses are performed to a high technical standard and are described in sufficient detail.

4. Conclusions are presented in an appropriate fashion and are supported by the data.

5. The article is presented in an intelligible fashion and is written in standard English.

6. The research meets all applicable standards for the ethics of experimentation and research integrity.

7. The article adheres to appropriate reporting guidelines and community standards for data availability.

Author Response

Thank you for your kind review.

Reviewer 2 Report

In the manuscript by Lee et al, Stability enhancement of freeze-dried gelatin/alginate coacervates for bFGF delivery. In this manuscript, authors have tried to solve he instability problems associated with bFGF delivery for chronic wound. Although manuscript tried to solve the instability problems using freeze drying technology in which we can store for long-term in powder state. However, authors need to address the following comments/ suggestions before publication.

-In introduction, authors need to explain more about freeze drying technology and how this technology gained tremendous interest for the preservation of drug delivery system, biomolecules, and other molecules, and role of lyoprotectants. For ref.

-https://www.mdpi.com/1999-4923/13/7/1052

-Section 2.2: Fabrication of Gelatin/alginate coacervates with or without cryoprotectant. In this section what is the ratio of lyoprotectants (Trehalose and Mannitol) and surfactant (Tween 80) added is not clear. This is because Tween 80 surfactant is not supposed to be in the formulation above 0.5%. Please clarify this.

-As residual water content in freeze dried product plays major role in long-term stability of the product. Did authors tried to quantify the residual water content in freeze dried product? This need to be performed.

-Glass transition temperature also very important to understand the transition phase of the product at that storage temperature. So, it is highly recommended to determine the glass transition temperature of the freeze-dried product using DSC.

-Its better to put the list of abbreviations and their compositions before introduction.

-  In the figure captions, it is better to explain more in details regarding number of replicates and if there is significant difference among control.

-In figure 4, there is clear scale bar, should be made clearer and also mention in the figure caption.

-All the figure captions need more explanation.

Author Response

We are grateful for the reviewer’s comments and suggestions. We have revised the manuscript to reflect the reviewer’s recommendation. Our point-by-point responses to the Reviewer’s comments are shown in the attached file. 

Reviewer 3 Report

The manuscript investigated  Stability Enhancement of Freeze-dried Gelatin/Alginate Coacervates for bFGF Delivery” the findings in the paper are of significance in engineering. It is a meaningful work, but I think the manuscript still has alittle problems and needs to do more work to improve the quality of this manuscript to meet the standards for publication in Materials.  Under the present state, I therefore suggest a minor revision of the manuscript. I have added a detailed list of comments below: they should be taken into account by the authors when reworking the manuscript.

2. The conclusion is too short and concise.

2. The following reference could support the discussion of this sentence.

Introduction of magnetic and supermagnetic nanoparticles in new approach of targeting drug delivery and cancer therapy application, ZM Avval, L Malekpour, F Raeisi, A Babapoor… - Drug metabolism reviews, 2020.

Bioactive agent-loaded electrospun nanofiber membranes for accelerating healing process: A review

SM Mousavi, ZM Nejad, SA Hashemi, M Salari… - Membranes, 2021

3- The English language must be carefully revised by a Native english Speaker.

4- References should be carefully rewritten.

5- The quality of Figure 2 is poor to be corrected.

6- Methodology. Examining the records you need to explain more.

Author Response

(The authors gave the same response as above.)

Round 2

Reviewer 2 Report

Authors have addressed the comments/suggestions, recommended to accept.